# Damage Identification of Stay Cables Based on a Small Amount of Deflection Monitoring Data

**Yanxiao Yang** [1,*] **and Mubiao Su** [2]

1    School of Civil Engineering, Shijiazhuang Tiedao University, Shijiazhuang 050043, China
2    Structural Health Monitoring and Control Institute, Shijiazhuang 050043, China
*    Correspondence: 220190113@student.stdu.edu.cn; Tel.: +86-18033704187

**Abstract:** For the problem of cable damage identification in cable-stayed bridges, we have presented a method for identifying cable damage based on deflection monitoring data from a small number of measurement points. We first describe the method to reduce the number of measurement points. We analyzed the distributional characteristics of the deflection difference before and after cable damage in cable-stayed bridges with optimized measurement points. The first derivative of the deflection difference is transformed by a wavelet transform to identify the location of the damaged cable. Then, the Kriging proxy model with exponential and deflection differences is established. The objective function is constructed from the residual deflection difference formed by the deflection difference and the measured deflection difference. With the particle swarm optimization algorithm, the damage parameters in the surrogate model are modified to minimize the objective function, and the damage to the cables is then identified. It is shown that the location of the damaged cable can be identified from the deflection data of a small number of measurement points with small error. The degree of damage can be accurately determined using the surrogate model.

**Keywords:** stay cable; damage identification; deflection difference; cable-stayed bridge; Kriging agent model

## 1. Introduction

The cable is the main mechanical component of the cable-stayed bridge. During the service period, due to environmental erosion and dynamic load, it is prone to wire breakage or other forms of damage [1–3], which leads to changes in cable force and thus changes the alignment of the main girder, even affecting the operation safety of the bridge in serious cases. During the lifetime of a cable-stayed bridge, how to master the damage condition of the cables (including the location and degree of the damage), scientifically evaluate the health status of the cables, provide timely health information of the cables for the management and maintenance department of the cable-stayed bridge, and determine whether the cables' replacement is needed to provide a decision basis have been popular and difficult issues in the field of bridge engineering at home and abroad for a long time. The damage identification methods for a cable (hanger) can be divided into two categories.

One of the cable damage identification methods is to directly detect the damage of cables. After the cable is damaged, the location and size of the damaged area of the cable can be judged by detecting the magnetic leakage intensity from the surface of the cable, However, this is hard to accomplish [4]. Through acoustic emission technology, the received acoustic emission signals are used to study or evaluate the dynamic integrity of structural materials, and the uncertain location of cable fatigue crack leads to the difficult application of acoustic emission technology in cable damage monitoring [5].

The other method is indirect detection of cable damage. At present, the damage identification of cable is mainly carried out through dynamic parameters (modal frequency, modal mode shape, curvature mode and flexibility curvature, modal strain energy, acceleration). Natural frequency can reflect the health state of the bridge. After structural damage

occurs, the natural frequency will change accordingly. In the early stage of the development of structural damage identification, most studies on structural damage identification are carried out with the change in natural frequency as the starting point [6,7]. After slight damage occurred to the cable, the natural frequency was a small variable [8]; therefore, it was difficult to identify the damage of the cable through frequency change.

Zhang et al. [9] decomposed the dynamic response of the cable into evanescent wave component and propagating wave component, and judged the local damage of the cable by the reflection coefficient of the blanking wave. Ren et al. [10] used bridge deck strain combined with support vector machine to identify cable damage. Huang et al. [11] proposed a cable damage identification method based on Kalman filtering and co-integration.

However, the dynamic response of the bridges is prone to the interference of the external environment. Due to the complexities of the actual bridge structures and the field environment, as well as the incomplete measured data, the uncertainty of the structural dynamic responses leads to the unsatisfactory practical application effect of the existing research results.

The method based on static parameters has attracted the attention of scholars. After the cable is damaged, the damage cable force decreases while the nearby cable force increases. The damage of the cable can be identified through the change in cable force [12]. In practical engineering, the cable force testing methods include jack hydraulic method, pressure sensor measurement method, frequency method, magnetic flux detection method [13,14], etc. These detection methods will be more or less affected by various factors in the practical application process so that the accuracy will be reduced.

Compared with the dynamic response of the structure, the signal of the bridge deflection response is more stable, which can not only represent the overall performance of the structure, but also reflect the local stress state of the structure, and is less affected by the external environment interference. Many scholars have studied the method of cable damage identification based on the deflection data of the main beam. Taking an arch bridge as the research background, Wang et al. [15] deduced the relationship between the deflection difference of the tie beam and the change in the cable force of the damaged hangers by using the relevant principles of structural mechanics, accurately identified the location of the damaged hangers by using the third derivative of the deflection difference of the tie beam, and verified the reliability of the method with laboratory models. Taking an arch bridge as the background as well, Wang et al. [16] proposed a derrick damage identification method based on the influence matrix of the deflection difference of the tie beam in the arch bridge by deducing the relationship between the deflection difference of the tie beam and the change in cable force, and verified the feasibility of the method in the laboratory. Fang et al. [17] identified the damage of the stay cable by using the particularity of a few non-zero coefficients in the residual force and static load vector of the stay cable. Chen et al. [3] suggested a method for detecting hanger damage based on the fluctuation of the displacement difference between measuring points. Wang et al. [18] introduced one new method to localize and quantify the partial cable damage using the abnormal variation in temperature-induced girder deflection caused by cable damage.

For a cable-stayed bridge with hundreds of cables, installing sensors at the anchorage point of each cable to measure the beam deflection is costly and workload-heavy. How to obtain the deflection distribution curve of the girder through the deflection of a few monitoring points is an urgent problem to be solved in cable damage identification based on the deflection of the girder. This paper aims to explore a more practical and maneuverable new method for cable damage identification of cable-stayed bridges based on the deflection monitoring data of optimized measuring points.

## 2. Theory Related to the Reduction in Measurement Points

Taking a single-pylon cable-stayed bridge as an example, its span arrangement is 232 m + 75.4 m + 34 m + 28.6 m, and the total length of the bridge is 370 m. The structure is shown in Figure 1. The main span (232 m) of the river side of the bridge is a three-way

prestressed concrete girder with an open box section and a height of 2.2 m. The bank side span is a closed box girder and a solid girder segment. The first span (75.4 m) is a box section, and the second span (34 m) and the third span (28.6 m) are solid sections. The full width of the girder is 23.5 m, and the width of the bridge deck is 16 m. The cables of the whole bridge are made of galvanized low relaxation high strength parallel steel wire with a diameter of 7 mm. The cable is divided into seven types; the standard tensile strength of the cable is 1670 Mpa and the elastic modulus is $2.06 \times 10^5$ Mpa. The cables are arranged in an asymmetric form. There are 52 pairs of cables on the upstream and downstream sides. The main girder of the box is made of concrete marked C50. Concrete marked C40 is used for the girder of the solid section and the tower.

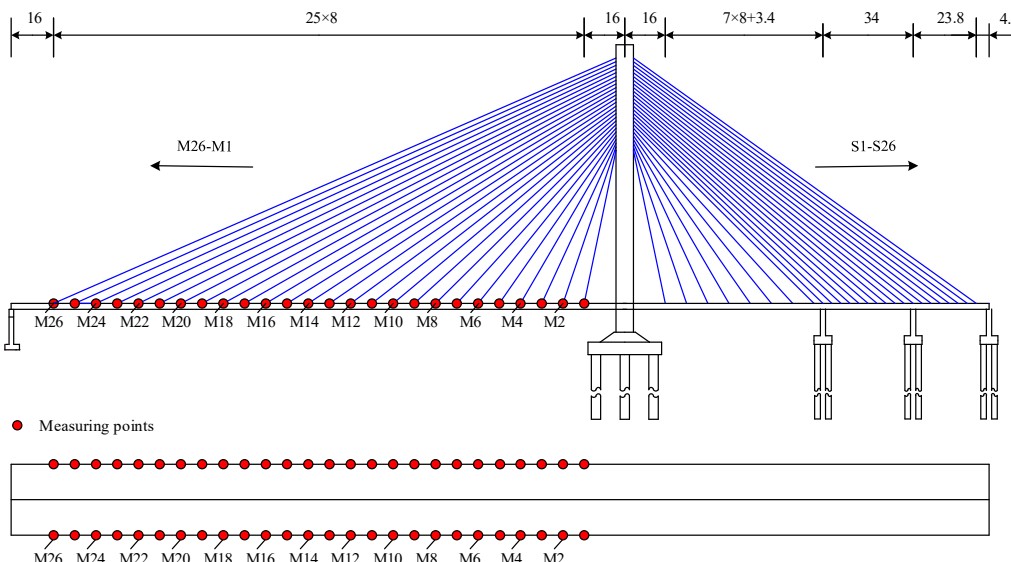

**Figure 1.** Cable number and deflection measuring point layout of a single−pylon cable-stayed bridge structure.

According to the design drawings and relevant data of the bridge, the three-dimensional finite element reference model of the cable-stayed bridge is established by ANSYS. Among them, beam188 beam element is used to simulate the main beam, solid65 solid element is used to simulate the pylon, and link10 element is used to simulate the stay cable. Corresponding constraints are imposed on the bottom of the main beam (corresponding position of pier), and the second stage dead load is equivalent to a uniform load on the bridge panel, ignoring the influence of pier on the deformation of the main beam.

After the cable is damaged, its stiffness decreases, which can be simulated by reducing its section area or elastic modulus [19]. The method of reducing the elastic modulus is adopted here. Assume that the damage index of cable $i$ is $\tau_i$. Then,

$$\tau_i = \frac{E_{di}}{E_{ui}} \tag{1}$$

where $E_{di}$ is the elastic modulus of cable $i$ after damage, and $E_{ui}$ is the elastic modulus of cable $i$ without damage.

### 2.1. Measuring Point Arrangement of Beam Deflection under Concentrated Load

As shown in Figure 2, the main beam of the cable-stayed bridge is equivalent to a foundation beam; it is a section AB taken from a section of the main beam of the cable-stayed bridge, which contains $n$ cables. The anchorage points of each cable and the main beam are numbered $1, 2, \cdots, k, \cdots, i, \cdots, n$ from left to right. The cable forces in the cable-stayed cables are $F_1, \cdots, F_k, \cdots, F_i, \cdots, F_n$, respectively. The angles between the stay cable and the main girder are $\alpha_1, \cdots, \alpha_k, \cdots, \alpha_i, \cdots, \alpha_n$, respectively.

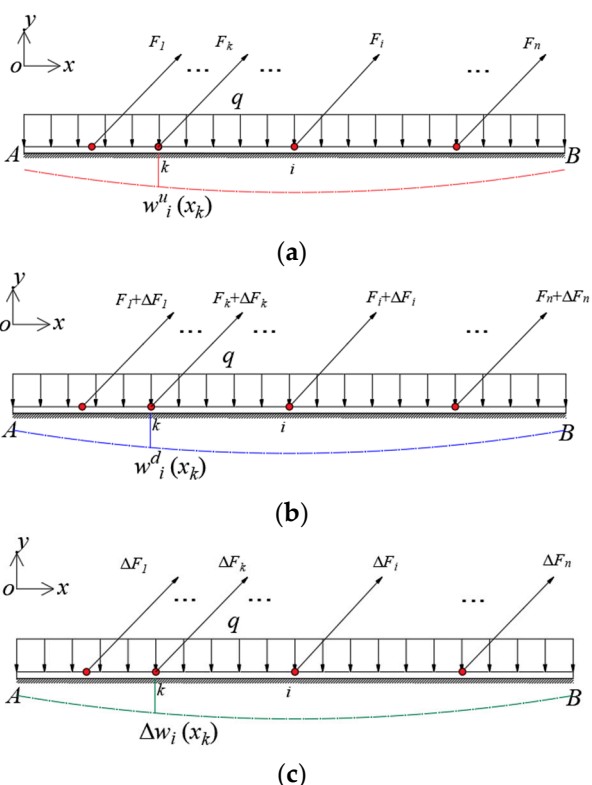

**Figure 2.** Stress analysis diagram of cable before and after damage: (**a**) in the healthy state; (**b**) after the cable damage; (**c**) before and after cable damage.

In Figure 2, $w_{k,i}^u(x)$ and $w_{k,i}^d(x)$ are the deflection at the anchor point of cable $k$ and the main beam, when one stay cable $i$ has been damaged at any position under the action of dead weight. The deflection difference of the beam before and after cable $i$ damage can be expressed using Equation (2).

$$\Delta w_k(x) = w_{k,i}^d(x) - w_{k,i}^u(x) \ (i,\ k = 1,\ 2,\ 3,\ \cdots,\ n) \tag{2}$$

where the subscript $i$, represents the position of the damage cable (can be multiple or single-cable damage), $n$ is the number of the anchorage points, $x_k$ is the anchorage points of each stay cable on the beam.

As shown in Figure 2a, the deflection of the beam at any section $k$ in healthy state can be expressed Equation (3).

$$w_{k,i}^u(x) = w_{k,i}^q(x) + \sum_{i=1}^n w_{k,i}^{F_i}(x), (i, k = 1, 2, 3, \cdots n) \tag{3}$$

where $w_i^q(x_k)$ is the deflection due to the effect of the uniform load $q$, $w_i^{F_i}(x_k)$ is the deflection at a cable force of $F_i$ when there is no damage to the cable.

When the damaged state of the cable is unknown (see Figure 2b), the deflection of the beam at any section $k$ can be expressed as in Equation (4).

$$w_i^d(x_k) = w_i^q(x_k) + \sum_{i=1}^n w_i^{F_i + \Delta F_i}(x_k), i, k = 1, 2, 3, \cdots n \tag{4}$$

where $w_i^{F_i + \Delta F_i}(x_k)$ is the deflection at a cable force of $F_i + \Delta F_i$ when the cable is damaged.

As shown in Figure 2c, when the cable $i$ is damaged, the force of the cable decreases, the nearby cable force of the nearby cable increases and the value is small; in addition, the

other cable's force is essentially unchanged and can be ignored. The deflection difference of the beam before and after cable damage can be expressed as in Equation (5).

$$\Delta w_i(x_k) = w_i^d(x_k) - w_i^u(x_k) = \sum_{i=1}^{n} w_i^{\Delta F_i}(x_k), i = 1, 2, 3, \cdots n \tag{5}$$

That is, the deflection difference $\Delta w_i(x_k)$ is caused by the change in the cable force of the damage cable $i$.

As shown in Figure 3, according to Winkle's hypothesis [20], the settlement of any point on the surface of a foundation is proportional to the pressure exerted on that point per unit area.

$$EId^4\Delta w(x_k)/dx^4 = -k\Delta w(x_k) + q \tag{6}$$

where $E$ is the elastic modulus of the main beam, $I$ is the bending inertial moment of the main beam, $k$ is the elastic coefficient of the main beam.

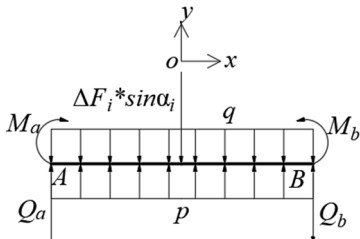

**Figure 3.** Microsegment diagram of girder subjected to force.

Under the action of concentrated force, the solution of Equation (6) is

$$\Delta w(x_k) = \frac{\Delta F_i sin\alpha_i \gamma}{2k} e^{-\gamma x_k}(\cos(\gamma x_k) + \sin(\gamma x_k)) \tag{7}$$

where $\gamma$ is the elastic eigenvalue of the beam, $\gamma = \sqrt[4]{k/4EI}$.

The deflection difference of the beam less than 0 can be obtained through symmetry. As shown in Figure 4, the deflection distribution curve of the beam takes the point of concentrated force as the symmetric point, and the graph on the left ($x < 0$) can be obtained symmetrically, as shown in Equation (8).

$$\Delta w(x_k) = \begin{cases} \frac{\Delta F_i sin\alpha_i \gamma}{2k} e^{-\gamma x_k}(\cos(\gamma x_k) + \sin(\gamma x_k)), \; x \geq 0 \\ \frac{\Delta F_i sin\alpha_i \gamma}{2k} e^{\gamma x_k}(\cos(\gamma x_k) - \sin(\gamma x_k)), \; x < 0 \end{cases} \tag{8}$$

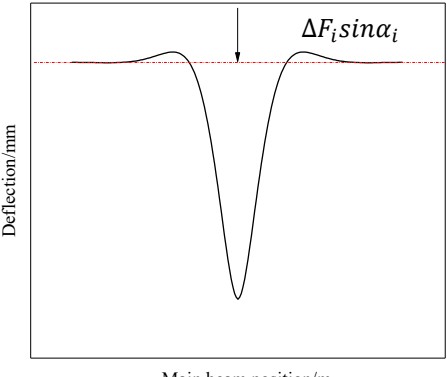

**Figure 4.** Schematic diagram of beam deflection curve under concentrated force.

According to Equation (8) and Figure 4, the beam deflection achieves its maximum value at the location where the concentrated force acts (the anchorage point of the damaged cable and the main beam). The deflection value at infinity tends to 0.

When the measurement point is close to the central force point, the beam deflection value is large and the deflection variation in the neighboring measurement points is large; therefore, the measurement point should be chosen with a small interval.

When the measurement point is far from the central force point, the deflection value of the beam is small and the variation in the deflection of adjacent measurement points is small so that the measurement point interval can be increased appropriately when the measurement point is chosen.

As shown in Figure 5, the previous cable damage identification method needs to obtain the deflection at the anchorage point of each cable and the main beam, with a total of 26 measuring points. The concentrated force can be applied to the anchorage point of cable M19. The selected measuring points are shown in Figure 5b. Compared with the measurement points before the reduction, only 14 measuring points are needed.

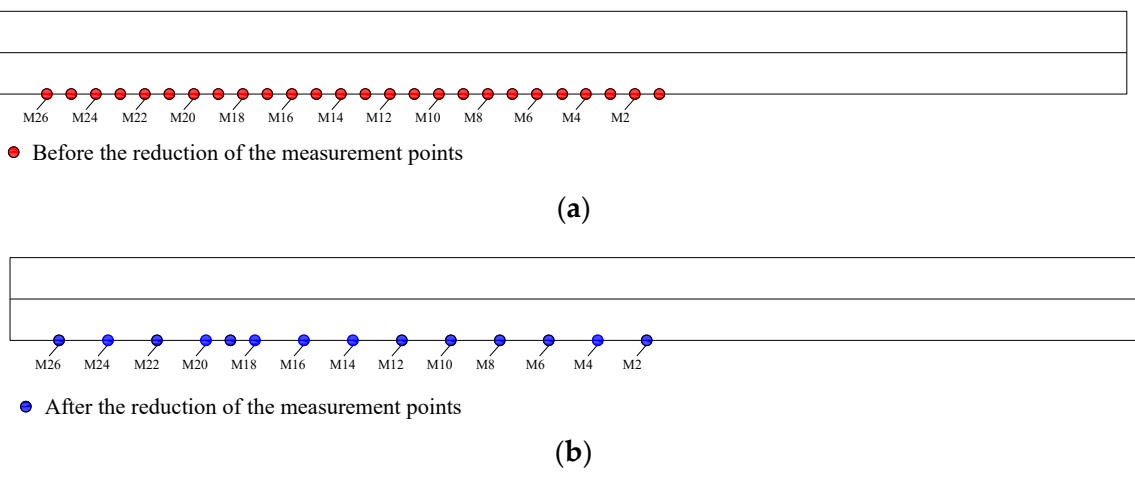

**Figure 5.** Methods for reducing the number of measurement points: (**a**) Before the reduction; (**b**) After the reduction.

*2.2. Calculation of the First Derivative of the Deflection Difference*

Find the first derivative of Equation (8):

$$\frac{d\Delta w(x_k)}{dx_k} = \begin{cases} -\frac{\Delta F_i \gamma^2 sin\alpha_i}{k} e^{-\gamma x_k} \sin(\gamma x_k), x \geq 0 \\ \frac{\Delta F_i \gamma^2 sin\alpha_i}{k} e^{\gamma x_k} \sin(\gamma x_k), x < 0 \end{cases} \tag{9}$$

According to Equation (9), when $x = 0$ (position of cable force change), the first derivative of deflection difference $\frac{d\Delta w(x_k)}{dx_k} = 0$. Therefore, the position of damaged cable can be judged according to the zero position of the first derivative distribution curve of deflection difference.

*2.3. Characteristic Analysis of Beam Deflection Difference Distribution Curve under Concentrated Force*

2.3.1. Analysis of Distribution Curves of Beam Deflection Difference before and after Single-Cable Damage

It is assumed that the single cable M16 has different degrees of damage (10%, 20%, 30% and 40%, respectively) when the concentrated force acts on the anchorage point of cable M19. According to the method described in Section 2.1, the deflection data of some measuring points (M26, M24, M22, M20, M19, M18, M16, M14, M12, M10, M8, M6, M4, M2, M1) are obtained. The cubic spline interpolation function is used to encrypt the deflection

data points and obtain the deflection values at the anchorage points of each cable and main beam. Figures 6 and 7 show the deflection and deflection difference distribution curves of measurement points before and after the reduction in the measurement points when single cable M16 has different degrees of damage.

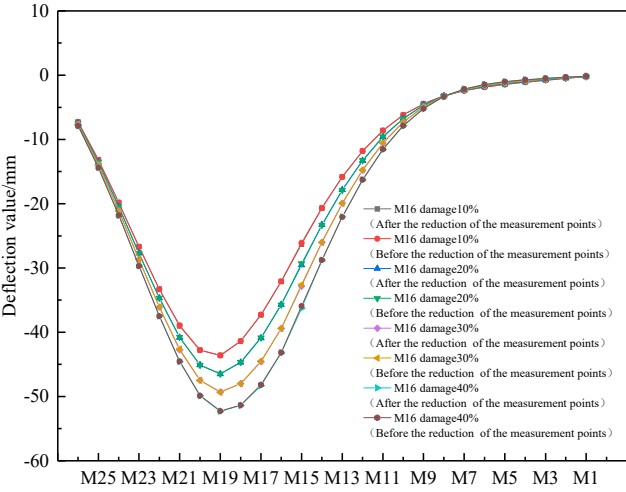

**Figure 6.** Comparison of deflection distribution curves before and after the reduction in the measurement points in the case of single-cable damage (M16 has different degrees of damage).

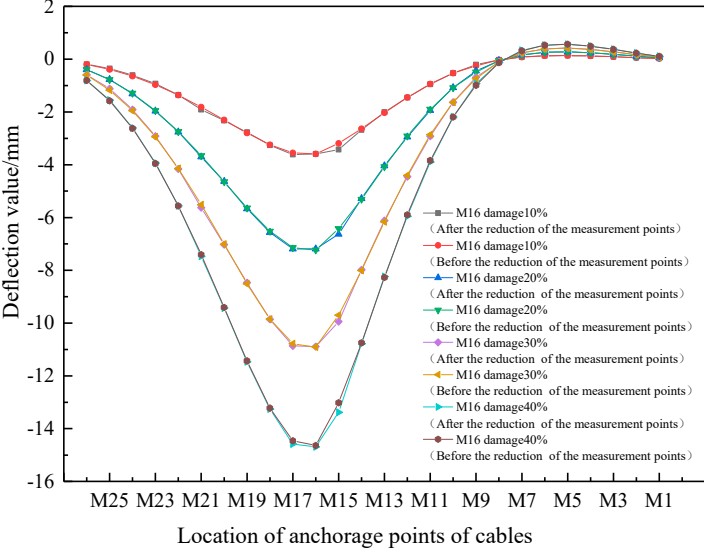

**Figure 7.** Comparison of deflection difference distribution curves before and after the reduction in the measurement points in the case of single-cable damage (M16 has different degrees of damage).

As can be seen from Figure 6, the deflection of the girder at the position where the concentrated force acts is the largest, and the deflection distribution curve of the girder obtained through the reduction in measuring points almost coincides with that before the reduction in the measurement points (the actual deflection distribution curve), with a maximum error of only 0.89%.

As can be seen from Figure 7, the girder deflection difference distribution curve obtained through the reduction in the measurement points of measuring points almost coincides with that before the reduction in the measuring points (actual deflection difference distribution curve) with no difference, and the maximum is only 2.76% (located at the anchorage point of cable M15 and main beam). When only cable M16 is damaged, the deflection difference distribution curve of the side girder where the damaged cable is

located is convex downward in the area near the anchorage point of the damaged cable, and the peak value (sharp point) appears at the anchorage point of the damaged cable, while the deflection difference corresponding to the anchorage point of each cable of the side girder where the non-damaged cable is located is very small (the calculation results are omitted). As the damage to the cable increases, the peak of the beam deflection difference distribution curve at the anchorage points of the damaged cable increases.

### 2.3.2. Analysis on Distribution Curve of Deflection Difference before and after Double-Cable Damage

It is assumed that the damage to both cables is different (see Table 1) when the concentrated force acts on the anchorage point of cable M19. Figure 8 is the comparison between the deflection distribution curve obtained after the reduction in the measurement points and the (actual) deflection distribution curve before the reduction in the measurement points when the double cable is damaged in the working condition listed in Table 1. Figure 8 is a comparison between the deflection distribution curve obtained after the reduction in the measurement points and the (actual) deflection difference distribution curve before the reduction in the measurement points when the double cable is damaged in the working condition listed in Table 1.

**Table 1.** Double cable damage case.

| Damage Type | Damage Cable | Damage Degree | Index of Damage |
| --- | --- | --- | --- |
| Case 1 | M7, M16 | 10%, 15% | 0.9, 0.85 |
| Case 2 | M7, M16 | 20%, 15% | 0.8, 0.85 |
| Case 3 | M7, M16 | 30%, 15% | 0.7, 0.85 |
| Case 4 | M7, M16 | 40%, 15% | 0.6, 0.85 |

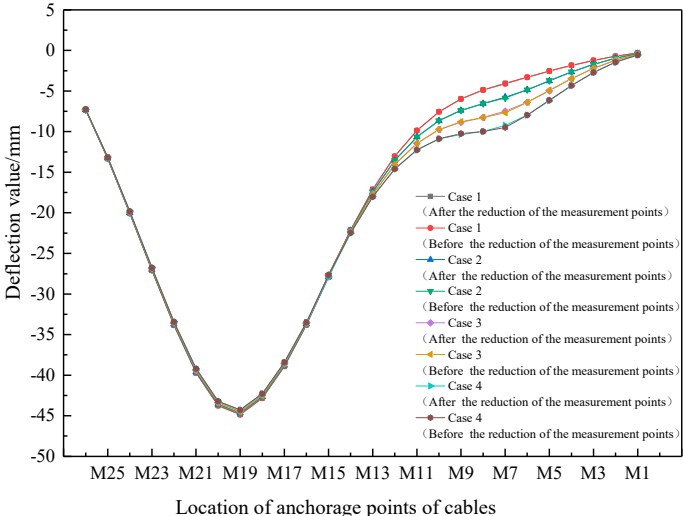

**Figure 8.** Comparison of beam deflection distribution curves before and after the reduction in the measurement points in the case of double-cable damage (different damage conditions).

As can be seen from Figure 8, when the double cables are damaged under the action of concentrated force, the girder deflection distribution curve after the reduction in the measuring points almost coincides with that before the reduction in the measurement points (the actual deflection distribution curve), with a maximum of only 2.51%.

As can be seen from Figure 9, the deflection difference distribution curve of girder before and after the reduction in the measurement points is essentially the same, with no difference, and the maximum is only 3.41% (located at the anchorage point of cable M7). After the damage of the double cable (M7 and M16), the deflection difference distribution curve of the side girder where the damage cable is located is convex downward in the area

near the anchorage point of the damage cable (M7 and M16), and a peak (point) appears. When the damage degree of one cable (such as M16) remains unchanged, while the damage degree of the other cable (such as M7) increases, the peak value corresponding to the beam deflection difference distribution curve at the anchorage point of M16 is essentially unchanged, while the peak value corresponding to the anchorage point of M7 increases with the increase in the damage degree.

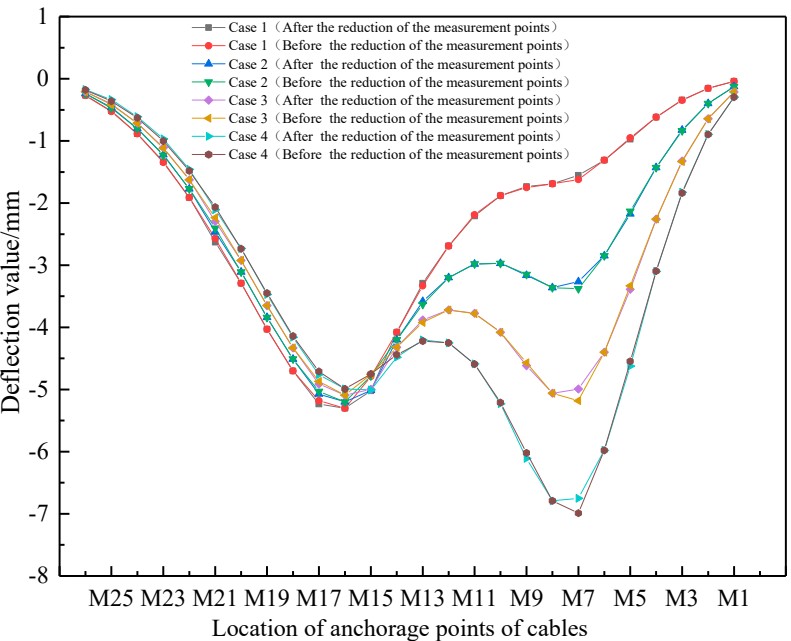

**Figure 9.** Comparison of beam deflection difference distribution curves before and after the reduction in the measurement points in the case of double-cable damage (different damage conditions).

## 3. Damage Location Identification Method of Stay Cable Based on First Derivative Wavelet Transform of the Girder Deflection Difference

### 3.1. Location Identification

Using the singularity detection feature of the wavelet transform, the first derivative of the girder deflection difference before and after cable damage is transformed by one-dimensional continuous wavelet transform [21].

$$W_f(a, b) = \frac{1}{\sqrt{a}} \int_{-\infty}^{+\infty} \Delta w'_{ij}(x_k) \overline{\psi} \left( \frac{x_k - b}{a} \right) dx \tag{10}$$

where $\Delta w'_{ij}(x)$ is the first derivative of deflection difference of the girder before and after the cable $i$ and $j$ damage (it can be double-cable damage or single-cable damage).

The Mexican hat Mexh wavelet formed by the second derivative of the Gauss function is chosen as follows:

$$\psi(t) = \frac{-1}{\sqrt{2\pi}} \left( 1 - t^2 \right) e^{\frac{-t^2}{2}} \tag{11}$$

The function satisfies the requirements of wavelet permitability and compact support, and its vanishing moment is 2.

Taking working condition 4 listed in Table 1 as an example, the damage cable location identification method in the case of double-cable damage is studied. The cubic spline interpolation function (spline) is used to properly encrypt the original data of girder deflection difference before and after the cable M16 damage of 15% and M7 damage of 40% (working condition 4); then, the first derivative of the encrypted girder deflection difference data is obtained according to Equation (9). The scale a = 11 is selected, and the first derivative of deflection difference is transformed by one-dimensional continuous

wavelet according to Equation (10). Figure 9 shows the distribution curves of the first derivative of the deflection difference of the beam before and after the double-cable damage (M16 damage 15%, M7 damage 40%).

It can be seen from Figure 10 that the distribution curve of the wavelet coefficient after wavelet transformation changes monotonically from negative to positive twice through the 0 point. The two 0 points are located at the anchorage points of cable M16 and M7, which are the same as the preset position of the damage cable.

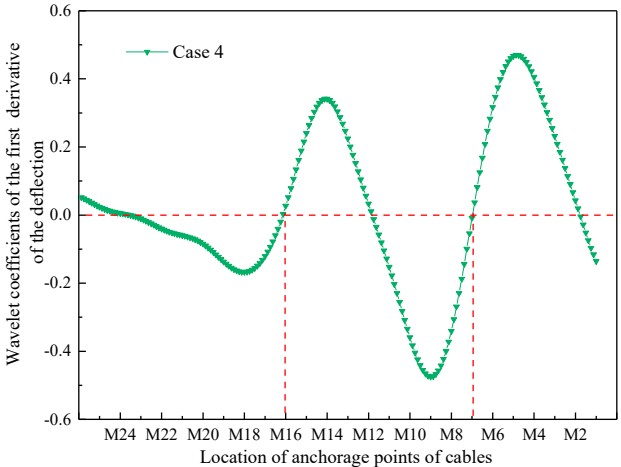

**Figure 10.** Wavelet coefficient distribution curves of the first derivative of beam deflection difference data encrypted before and after double-cable damage (M16 damage 15%, M7 damage 40%).

According to the above method, the damage cable position is identified for other preset conditions of single-cable damage and double-cable damage (results are shown in Figures 11 and 12).

As can be seen from Figures 11 and 12, the distribution curve of the wavelet coefficients changes from negative to positive across the X axis, and the location of the 0 point is the location of the damaged cable; in addition, the identification results are exactly the same as for the preset damaged cable location.

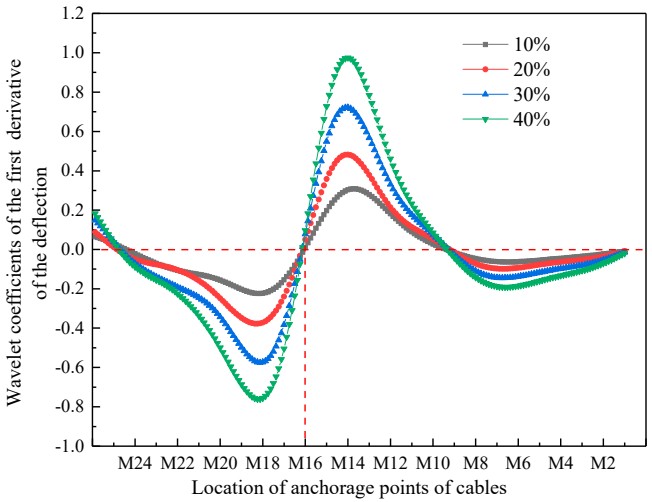

**Figure 11.** Wavelet coefficients distribution curves of the first derivative of beam deflection difference before and after cable M16 damage.

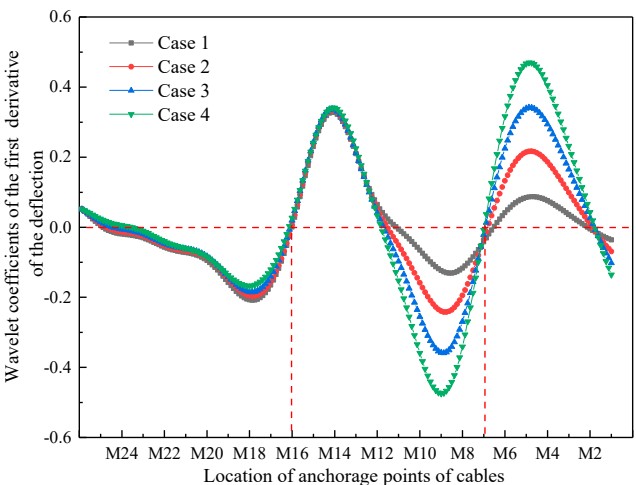

**Figure 12.** Wavelet coefficients distribution curves of the first derivative of beam deflection difference before and after cable M16 and M7 damage.

### 3.2. Damage Degree Identification

The variation in cable damage degree is random regardless of whether it is for single-cable damage or double-cable damage. With different cable damage degrees, the corresponding girder deflection variation and distribution law are also different. In order to identify the damage degree of a cable using numerical simulation, the calculation workload is too large. By using the Kriging proxy model, an approximate function can be constructed with limited sample data to predict the girder deflection corresponding to the damage degree of the cable so as to achieve the purpose of identifying the damage degree of the cable.

The Kriging proxy model is an interpolation technique for predicting unknown sample information based on the correlation of known sample information and the minimum variance criterion [22]. Let the response of an unknown vector $z = [z_1, z_2, z_3, \cdots z_n]^T$ be $Y = [y(z_1), y(z_2), y(z_3), \cdots y(z_n)]^T$, which can be predicted using the expression $\hat{y}(z)$ obtained via the superposition of the polynomial regression model $f^T(z)\alpha$ and the zero-mean normal distribution random function $\mathbf{g}(z)$.

$$\hat{y}(z) = \sum_{m=1}^{p} \alpha_m f_m(z) + \mathrm{g}(z) = f^T(z)\alpha + \mathbf{g}(z) \tag{12}$$

where the first part is the polynomial regression model about variable $z$, $\alpha$ is the regression coefficient, $\alpha = [\alpha_1\, \alpha_2 \ldots \alpha_p]^T$, $f = [f_1(z) f_2(z)\; \cdots\; f_p(z)]^T$; the second part $\mathbf{g}(z)$ is a random distribution with non-zero covariance and obeys the normal distribution $N(0, \sigma^2)$.

In order to judge whether the response predicted by the Kriging proxy model can meet the accuracy requirements, the square correlation coefficient and the square error criterion is generally used to evaluate its accuracy as shown in Equations (13) and (14).

$$SC = 1 - \frac{\sum_{i=1}^{N}\left[\hat{y}_i - y_i^b\right]^2}{\sum_{i=1}^{N}\left[y_i^b - \overline{y}\right]^2} \tag{13}$$

$$EISE = \frac{1}{N}\sum_{i=1}^{N}\left[\hat{y}_i - y_i^b\right]^2 \tag{14}$$

In Equation (14), $\hat{y}_i$ is the $i$-th component of the response vector predicted by the Kriging proxy model, $y_i^b$ is the $i$-th component of the actual response vector simulated by the numerical model, $\overline{y}$ is the mean value of $y_i$, and $N$ is the length of $\hat{y}$. In general, when $SC > 0.99$ and $EISE < 0.01$, it indicates that the response predicted by the proxy model meets the accuracy requirements; otherwise, it needs to be modified by adding criteria. In this

paper, the multi-point and multiple-point criterion is adopted to revise the Kriging agent model.

After the location of the damage cable is determined, the damage index of the damage cable is used as the parameter to be identified, and the Kriging proxy model is used for the quantitative identification of cable damage. It can be achieved through the following three steps:

1. The damage index of the damage cable is selected as the damage parameter, the uniform design method is used to set the sample points of the damage index, and the finite element model is used to simulate and calculate the deflection difference of the girder corresponding to the sample points (different damage degrees of the damage cable). Assume that cable *i* and cable *j* are damaged; the damage index of the damage cable is $z$, and the corresponding deflection difference of the girder is $y$.

$$
\begin{cases}
\mathbf{z} = \begin{bmatrix} \mathbf{z}_1 \\ \mathbf{z}_2 \\ \vdots \\ \mathbf{z}_k \\ \vdots \\ \mathbf{z}_n \end{bmatrix} = \begin{bmatrix} \tau_{1i} & \tau_{1j} \\ \tau_{2i} & \tau_{2j} \\ \vdots & \vdots \\ \tau_{ki} & \tau_{kj} \\ \vdots & \vdots \\ \tau_{ni} & \tau_{nj} \end{bmatrix} \\
\mathbf{y} = \begin{bmatrix} \mathbf{y}_1 \\ \mathbf{y}_2 \\ \vdots \\ \mathbf{y}_k \\ \vdots \\ \mathbf{y}_n \end{bmatrix} = \begin{bmatrix} \Delta w_{1i} & \Delta w_{1j} \\ \Delta w_{2i} & \Delta w_{2j} \\ \vdots & \vdots \\ \Delta w_{ki} & \Delta w_{kj} \\ \vdots & \vdots \\ \Delta w_{ni} & \Delta w_{nj} \end{bmatrix}
\end{cases}
\tag{15}
$$

where $n$ is the number of sample points, $\mathbf{z}_k$ is the damage index vector of the k-th sample point, $\mathbf{z}_k = \begin{bmatrix} \tau_{ki} & \tau_{kj} \end{bmatrix}^T$, $\tau_{ki}$ and $\tau_{kj}$ are the damage indexes of cable *i* and cable *j* corresponding to the *k*-th sample point, respectively, $\mathbf{y}_k$ is the deflection difference vector corresponding to the girder when the damage index is $\mathbf{z}_k$, $\mathbf{y}_k = \begin{bmatrix} \Delta w_{ki} & \Delta w_{kj} \end{bmatrix}^T$, $\Delta w_{ki}$ and $\Delta w_{kj}$ are the deflection difference at the anchorage point of cable *i* and cable *j* corresponding to the *k*-th sample point, respectively. Taking the working conditions listed in Table 1 as an example, the damage indexes of cable M16 and cable M7 ($\tau_{k16}$, $\tau_{k7}$) are selected as the parameters to be identified, and the variation interval of each parameter is set as (0–1). The uniform design method is used to design according to seven levels ($n = 7$). The values of each parameter are 0, 0.17, 0.33, 0.5, 0.67, 0.83 and 1.0, respectively. Seven groups of sample points, namely vector $z$, are designed using uniform design table $U_7(7^4)$ and its use table (see Table 2 for specific values). When a single cable is damaged, one of the parameters is set to a fixed value of 1. The finite element model is used to calculate the deflection difference vector y corresponding to the damage index sample points listed in Table 2.

2. The dace toolbox in matlab [23] is used to establish the functional relation between the deflection difference vector $\mathbf{y}_k$ and the damage index vector $\mathbf{z}_k$. The response surface between $\mathbf{y}_k$ and $\mathbf{z}_k$ is obtained by using the repoly0 function as the regression model (namely the Kriging proxy model). Figure 13 shows the response surface between $\tau_{k16}$, $\tau_{k7}$ and $\Delta w_{k7}$ (the deflection difference of M$_7$). Figure 14 shows the response surface between $\tau_{k16}$, $\tau_{k7}$ and $\Delta w_{k16}$ (the deflection difference of M$_{16}$). The accuracy requirements are satisfied by the checking calculation ($SC = 0.9998 > 0.99$, $EISE = 0.00187 < 0.01$). It can be seen from Figures 12 and 13 that every point falls on the response surface, indicating that the response surface is well-fitted and can be used instead of the finite element model for calculation.

3. Establish an objective function for the cable damage index to be identified, and then use the particle swarm optimization (PSO) algorithm [24] to find a set of data values to minimize the objective function; in addition, the corresponding data of the minimum value is the cable damage index. The deflection difference between the measured deflection difference of the girder before and after cable damage and the deflection difference predicted by the Kriging proxy model are compared, and the sum of the squares of the difference is taken as the objective function.

$$P_{obj} = min \sum_{i=1}^{N} (\overline{w}_i(z) - w_i(z))^2 \tag{16}$$

where $\overline{w}_i(z)$ is the *i*-th component of the deflection difference vector corresponding to point *z* estimated using Kriging's proxy model; $w_i(z)$ is the *i*-th component of the deflection difference vector actually measured; *z* is the vector formed by the damage index variable; *N* is the vector length.

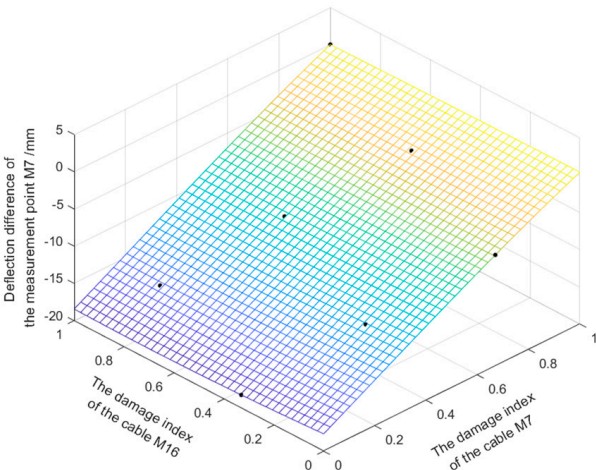

**Figure 13.** Kriging model between the deflection difference of $M_7$ and damage index ($\tau_{k16}$ and $\tau_{k7}$).

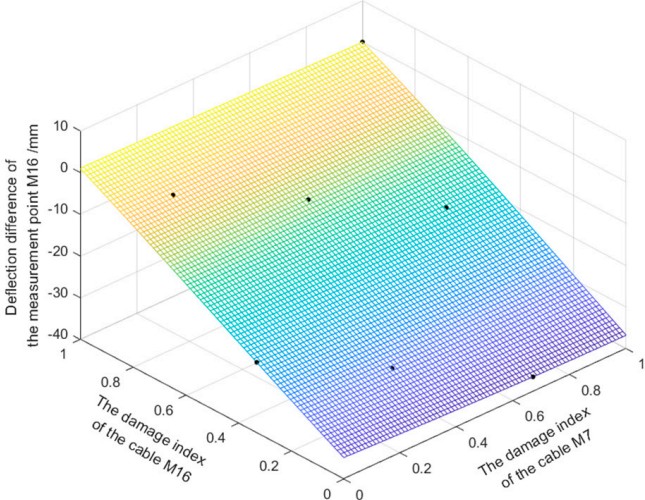

**Figure 14.** Kriging model between the deflection difference of $M_{16}$ and damage index ($\tau_{k16}$ and $\tau_{k7}$).

**Table 2.** Experimental design of Kriging proxy model.

| Damage Index | $\tau_{k7}$ | $\tau_{k16}$ |
|---|---|---|
| $z_1$ | 0 | 0.33 |
| $z_2$ | 0.17 | 0.83 |
| $z_3$ | 0.33 | 0.17 |
| $z_4$ | 0.5 | 0.67 |
| $z_5$ | 0.67 | 0 |
| $z_6$ | 0.83 | 0.5 |
| $z_7$ | 1 | 1 |

Set the number of ethnic groups to 2, the scale of ethnic groups to 100, the search dimension to 2, and the search scope to (0, 1). The identification result for the cable damage index can be obtained by adjusting the damage index in the surrogate model until the objective function reaches a minimum.

Table 3 identifies the deflection difference of the girder after the reduction in the measurement points, the deflection difference of the girder before the reduction in the measurement points and the preset value of the damage index.

**Table 3.** Damage index identification result table.

| Single or Double-Cable Damage | Actual Value | | Recognized Value (After the Reduction in the Measurement Points) | | Error (After the Reduction in the Measurement Points) | | Recognized Value (Before the Reduction in the Measurement Points) | | Error (Before the Reduction in the Measurement Points) | |
|---|---|---|---|---|---|---|---|---|---|---|
| | M7 | M16 | M7 | M16 | M7 | M16 | M7 | M16 | M7 | M16 |
| Case 1 | 0.9 | 0.85 | 0.9010 | 0.8404 | 0.11% | −1.13% | 0.8969 | 0.8403 | 0.35% | −1.14% |
| Case 2 | 0.8 | 0.85 | 0.8017 | 0.8413 | 0.21% | −1.02% | 0.7950 | 0.8412 | 0.62% | −1.04% |
| Case 3 | 0.7 | 0.85 | 0.7049 | 0.8423 | 0.70% | −0.91% | 0.6946 | 0.8420 | 0.77% | −0.94% |
| Case 4 | 0.6 | 0.85 | 0.6094 | 0.8427 | 1.56% | −0.86% | 0.5966 | 0.8424 | 0.57% | −0.89% |
| Damage 10% | - | 0.9 | - | 0.8920 | - | −0.89% | - | 0.8920 | - | −0.89% |
| Damage 20% | - | 0.8 | - | 0.7977 | - | −0.29% | - | 0.7965 | - | −0.43% |
| Damage 30% | - | 0.7 | - | 0.6938 | - | −0.88% | - | 0.6933 | - | −0.96% |
| Damage 40% | - | 0.6 | - | 0.5969 | - | −0.51% | - | 0.5963 | - | −0.62% |

Note: Error = (identified value − actual value)/Actual value ∗ 100%.

It can be seen from Table 3 that the identification value of the damage index obtained through the reduction in the measurement points is essentially the same as the actual value, and the error is small (the maximum is only 1.14%). It can be seen that the method of identifying the cable damage degree by using a small number of measurement points and based on the Kriging proxy model is more accurate.

## 4. Analysis of Influencing Factors of Cable Damage Identification

### 4.1. Influence of Noise on Damage Identification

4.1.1. Influence of Noise on Location Identification of Damage Cable

The girder deflection data before and after cable damage in this paper are obtained based on finite element numerical simulation, while the girder deflection data before and after cable damage in actual engineering should be field-measured data. When the beam deflection of cable-stayed bridge is detected, the test signal will be interfered with by various factors. In order to test the anti-noise performance of the above method, white Gaussian noise is added to the numerical simulation of beam deflection difference data, and then the damage location and damage degree are identified.

Taking single-cable damage and double-cable damage conditions as examples, white noise (signal-to-noise ratio of 20 dB) is added to the deflection difference data of the girder corresponding to each working condition to obtain the first derivative of the deflection difference of the girder. Then, the wavelet analysis is carried out on it. Figure 14 shows the

distribution curve of the first derivative of the deflection difference of the girder before and after cable damage under different damage conditions with the addition of white noise.

According to Figure 15, when a single cable is damaged, the damage cable is M16. In the case of double-cable damage, the damage cables are M16 and M7. It can be seen that noise does not affect the accuracy of damage cable location identification.

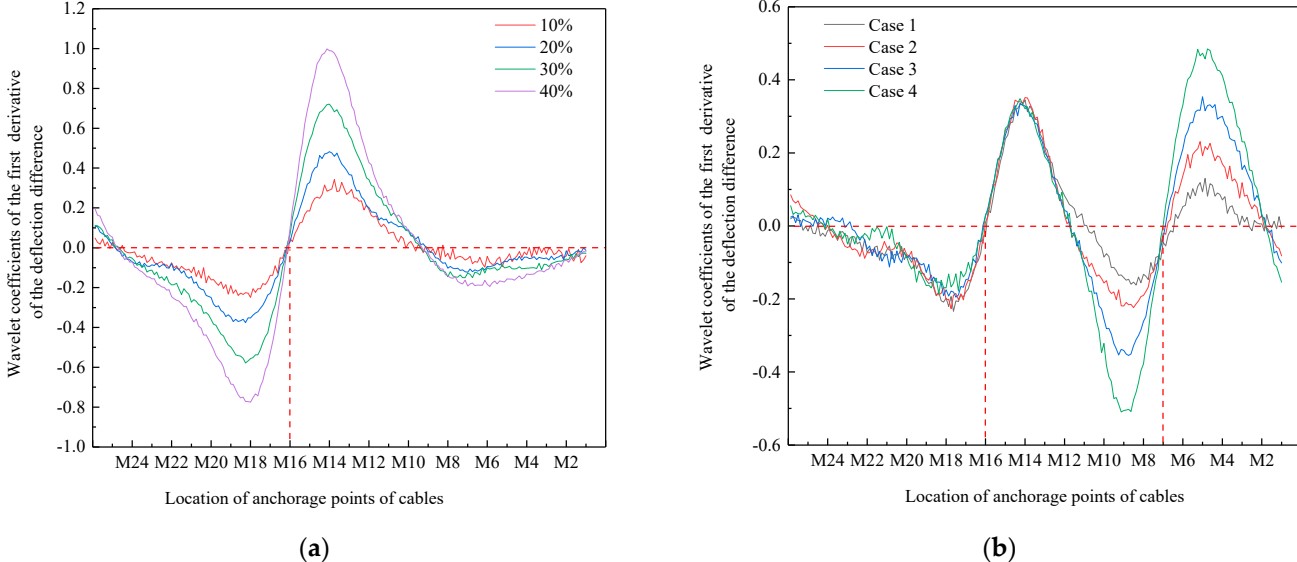

**Figure 15.** Wavelet coefficient distribution curves of the first derivative of beam deflection difference under different damage conditions after adding white noise. (**a**) Single-cable damage, (**b**) double-cable damage.

### 4.1.2. Influence of Noise on Damage Degree Identification

The deflection difference of the girder at the anchorage points of cable M7 and M16 after adding noise (instead of the measured deflection difference) is used as the basis to identify the damage degree of the cables. The damage index is re-identified according to the damage degree identification method, and the damage identification results of single-cable and double-cable can be obtained. The results show that the difference between the recognized damage index and the actual damage index (the preset damage index) is very small (the maximum error is 1.2%). It can be seen that the addition of noise will not affect the accuracy of identifying cable damage degree based on the Kriging proxy model.

### 4.2. Influence of the Location of Concentrated Load

#### 4.2.1. Location Identification

When the concentrated force is located at measuring points M4, M8, M13, M19, M21 and M24, the deflection of measuring points is optimized when the damage of cable M16 is 15% and the damage of cable M7 is 40%, and the damage cable location is identified according to the damage cable location identification method. Figure 16 is the wavelet coefficient distribution curve of the first derivative of the deflection difference of the girder when the concentrated force acts on different measuring points, and the legend represents the measuring point position of the concentrated force.

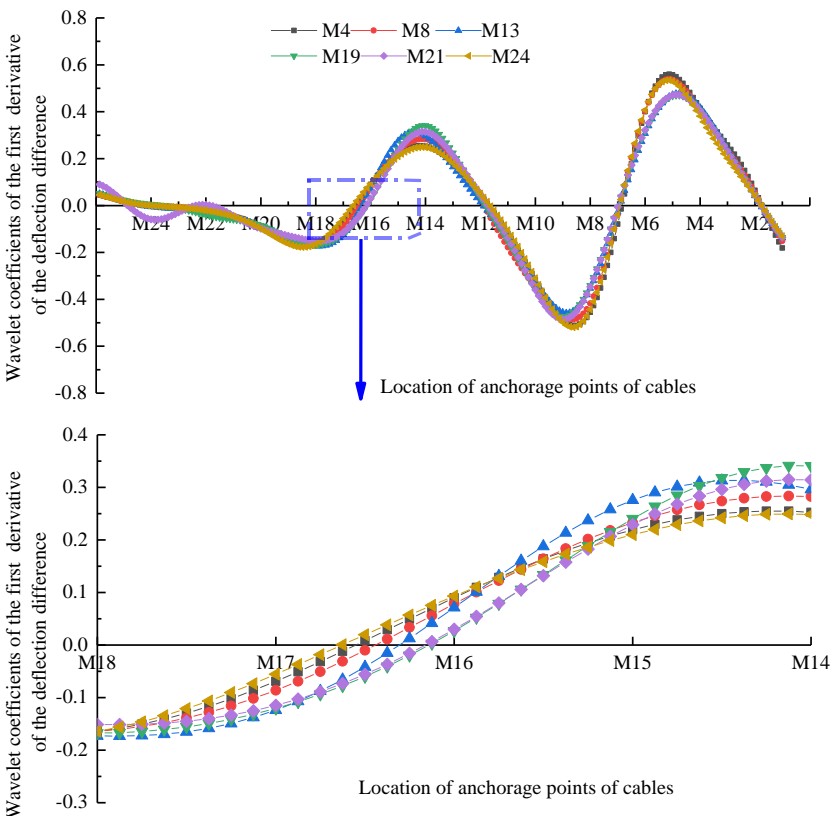

**Figure 16.** Distribution curves of the first derivative of the deflection difference of the girder when the concentrated force acts on different measuring points.

According to Figure 16, when the concentrated force acts on different measuring points, the wavelet coefficient of the first derivative of the deflection difference of the girder changes monotonically from negative to positive twice through the 0 point. One 0 point is located at the anchorage point of cable M7, and the other 0 point is located near the anchorage point of cable M16. When the acting position of the concentrated force is close to the cable tower (such as M4, M8) and the support (such as M24), the 0 point is far from the anchorage point of cable M16, and the accuracy of damage cable location identification decreases. It can be seen that the location of the concentrated force has an impact on the identification accuracy of the damage cable position. The reason is that when the location of the concentrated force is far away from the span, the impact on the deflection of the girder is small, and the error of the damage location identification is large.

### 4.2.2. Damage Degree Identification

The deflection difference of the girder (instead of the deflection values measured in real bridges) obtained when the concentrated force acts on different measuring points is used as the basis to identify the damage degree of the cable. The damage index is identified according to the damage degree identification method, and the results of four kinds of double-cable damage identification in working conditions can be obtained. The results show that when the concentrated force acts near the measuring point of the tower, the discernable damage index value is significantly different from the actual damage index (the preset damage index) value (the maximum error is 3.76%). It can be seen that the location of the concentrated force will affect the accuracy of identifying the cable damage degree based on the Kriging proxy model. To sum up, the point of operation of concentrated force should be selected in the mid-span as far as possible.

### 4.3. Influence of Bending Stiffness of Beam Body

4.3.1. Location Identification

With the extension of the service time, the bending stiffness (EI) of the cable-stayed bridge may degrade as a whole. The stiffness' degradation is simulated by the change in the elastic modulus of the beam (the moment of inertia of the section remains unchanged). Assume that the stiffness of the main beam decreases by 40%. Damage conditions are preset for verification, as shown in Table 1 (double cable). The identification results are shown in Figure 17.

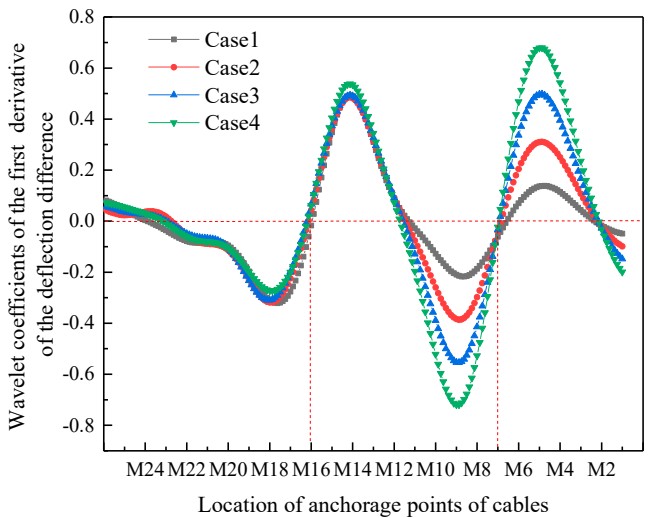

**Figure 17.** Identification results for considering stiffness degradation.

According to Figure 17, in the case of double-cable damage, the damage cables are M16 and M7. It can be seen that the stiffness' degradation does not affect the accuracy of damage cable location identification. However, the wavelet coefficient corresponding to the position of the damage cable increases compared with before, which is caused by the increase in the deflection difference of the beam body after the stiffness degradation of the main beam.

4.3.2. Damage Degree Identification

The deflection difference of the beam at the anchorage point of cable M7 and M16 after the stiffness of the main beam decreases is used as the basis to identify the damage degree of cable. The damage index is re-identified according to the damage degree identification method, and the damage identification results of double-cable can be obtained. The results show that the difference between the recognized damage index and the actual damage index (the preset damage index) is very small (the maximum error is 1.5%). It can be seen that the when the stiffness of the main beam decreases, the accuracy of damage degree identification is improved.

### 4.4. Influence of Model Error

Model error, for example, unintentional eccentricities of the load (producing torsion) or an imperfect symmetry of the geometric or mechanical distribution of masses or stiffnesses may occur errors. It can also compromise the ability of the proposed method to identify cable damage. When the cable is damaged, it only affects the deflection of the damaged side of the cable, and the model error will increase or decrease the deflection difference of the beam body without affecting the location identification of the damaged cable. As the damage degree identification is related to the deflection difference, it has an impact on the damage degree identification; however, the error is small.

## 5. Experimental Verification

### 5.1. Introduction to Experimental Model

As shown in Figure 18, the laboratory model bridge is not designed in strict accordance with the similarity ratio of an actual single-pylon cable-stayed bridge; however, it can simulate the basic mechanical characteristics of a single-pylon cable-stayed bridge. The length of the cable-stayed bridge model is 3.6 m, the height of the tower is 1.6 m, and the distance between the cable-stayed cables on the bridge is 0.3 m. There are 20 cable-stayed cables in total, and the thickness of the bridge panel is 4 mm.

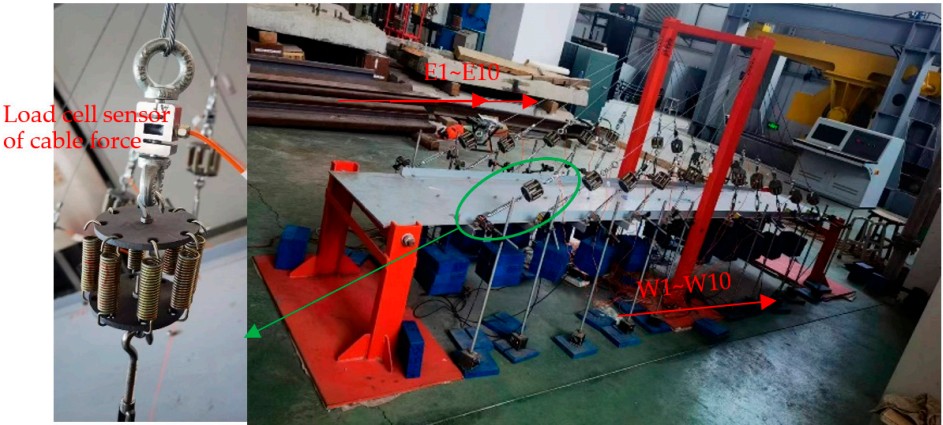

**Figure 18.** Experimental model of cable-stayed bridge.

To accurately control the preset degree of damage, the cable was specially designed in this model. The cable is mainly composed of four parts in series including a wire rope segment with a diameter of 3 mm, load cell for cable force, spring segment (consisting of eight springs with the same stiffness in parallel), and small flanges for adjusting cable force (see Figure 19). The test bridge was instrumented with a dense array of sensors, including eighteen displacement sensors with an accuracy of 0.001 mm, and twenty load cell sensors for cable force. The diagram of the sensors are shown in Figure 19, illustrating the locations of the deflection sensors. The measurement point of the west side is W1–W10, and that of the east side is E1–E10 (see Figure 20).

### 5.2. Identification of Cable Damage

Six damage cases were simulated using the laboratory test model, including both single-cable damage and double-cable damage types. The specific damage cases are shown in Table 4.

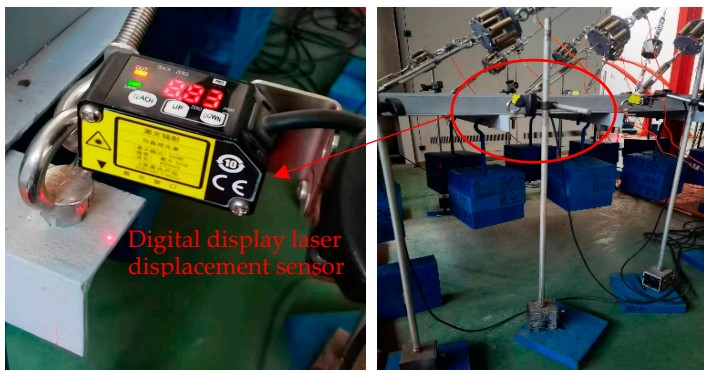

**Figure 19.** Beam deflection measuring device.

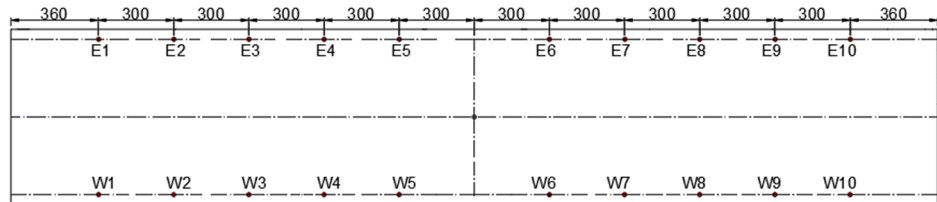

**Figure 20.** Measuring points arrangement.

**Table 4.** Damage cases.

| Damage Type | Damage Cable | Damage Degree | Index of Damage |
| --- | --- | --- | --- |
| Case 1 | W-4 | 22.4% | 0.776 |
| Case 2 | W-4 | 43.8% | 0.562 |
| Case 3 | W-4 | 69.9% | 0.301 |
| Case 4 | W-3 & W-6 | 26.0%, 41.7% | 0.74, 0.583 |
| Case 5 | W-3 & W-6 | 46.1%, 41.7% | 0.539, 0.583 |
| Case 6 | W-3 & W-6 | 83.2%, 41.7% | 0.168, 0.583 |

5.2.1. Location Identification

The cable force of each cable was adjusted to 400 N, and the state can be considered as the completed state of the bridge. As shown in Figure 21 vehicles are added to the cable-stayed bridge to simulate concentrated loads. The front wheel corresponds to the cable (W-5). The mass of the vehicle and counterweight was summed to 47.17 kg, and the line of the main girder at this time was measured as the line under concentrated load in a healthy state. Then, the line of the main girder was simulated when the damage listed in Table 4 (Case1–Case3) occurred in cable W-4. Figure 22 shows the deflection values of the main beam under concentrated load, in the healthy state and after the W-4 damage, of 22.4%, 43.8% and 69.9% (Case1–Case3).

Following the location identification method described in Section 3, the damage location identification results were obtained, as shown in Figure 23. From Figure 23a, the wavelet coefficient distribution curve of the first derivative of the deflection difference intersects the null point at the anchorage of cable W-4 and the main beam, the same as the preset damage location. From Figure 23b, the damage cable was W-3 and W-6, the same as the preset damage location.

5.2.2. Damage Degree Identification

The results of the damage identification are shown in Figure 24.

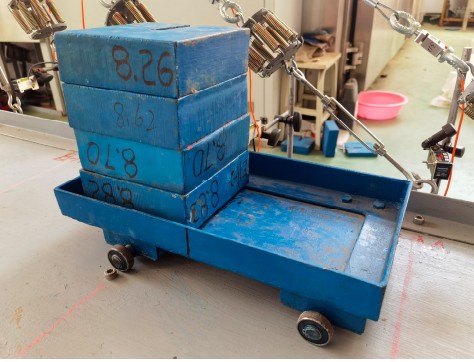

**Figure 21.** Vehicle loading diagram.

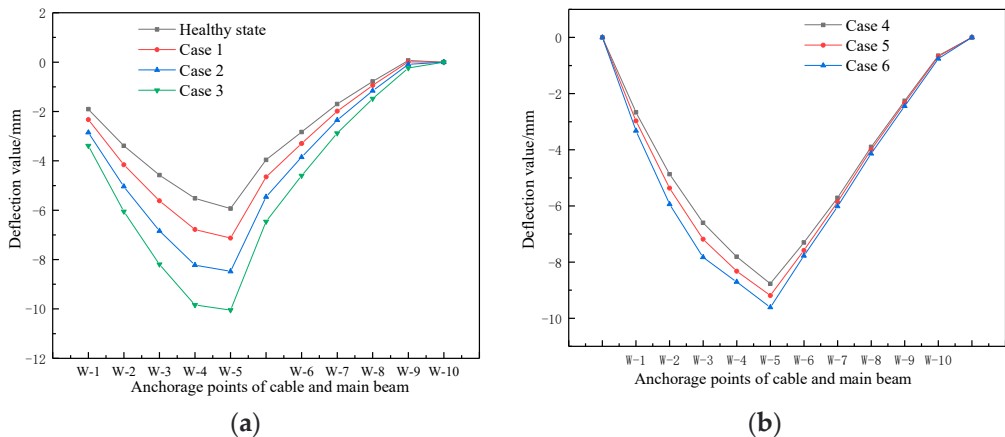

**Figure 22.** Deflection distribution curves of main beam under concentrated load. (**a**) Single-cable damage; (**b**) double-cable damage.

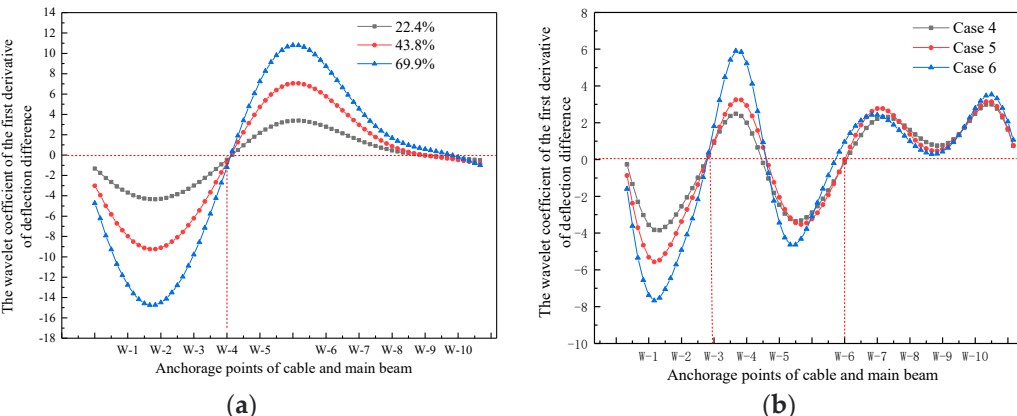

**Figure 23.** Damage location identification results. (**a**) Single-cable damage; (**b**) Double-cable damage.

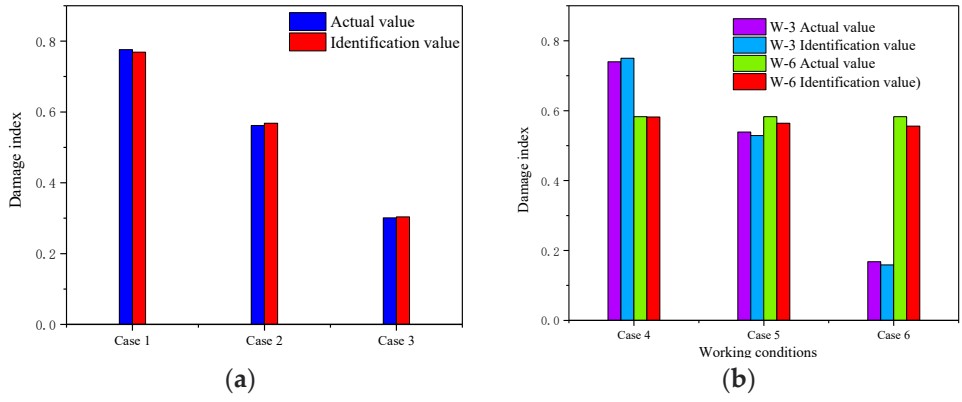

**Figure 24.** Damage degree identification results. (**a**) Single-cable damage; (**b**) Double-cable damage.

As can be seen from Figure 24, the damage identification index is essentially the same as the preset value, which indicates that the proposed damage identification method is feasible.

### 5.3. Influence of Noise on Damage Identification

Following the location identification method described in Section 3, after adding noise, the location identification of damage cable results are shown in Figure 25.

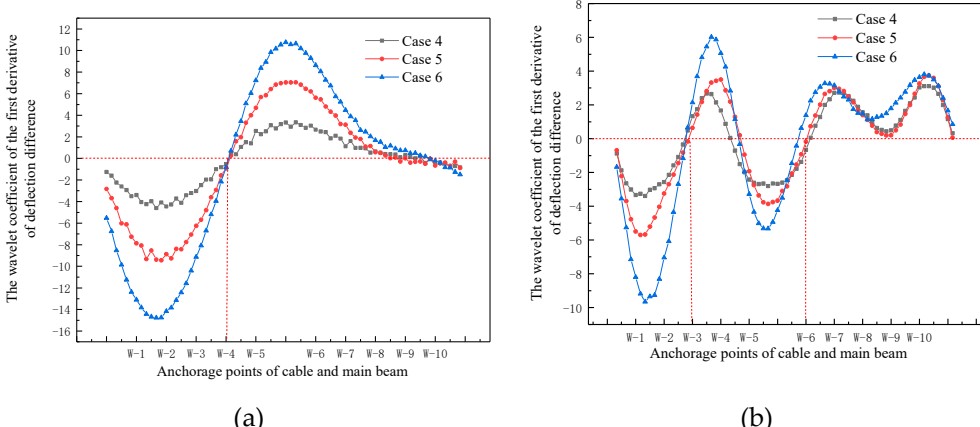

**Figure 25.** Cable damage identification results (after adding noise). (**a**) Single-cable damage; (**b**) Double-cable damage.

As shown in Figure 25, the identified damage location is the same as the preset damage rope location.

After adding noise, the results show that the difference between the recognized damage index and the actual damage index (the preset damage index) is very small (the maximum error is 2.3%). It can be seen that the addition of noise will not affect the accuracy of identifying cable damage degree based on the Kriging proxy model.

## 6. Conclusions

1. A small amount of deflection data can be used to identify the location and degree of cable damage. The wavelet coefficient distribution curve of the first derivative of the girder deflection difference before and after the cable damage monotonically changes from negative to positive from left to right on either side of the anchoring point where the damaged cable passes through the 0 point.
2. The deflection difference distribution curve of the girder obtained using a small amount of measurement points almost coincides with that obtained with the original number of measurement points. The first derivative of the deflection difference of the girder before and after the reduction in the measurement points of the measurement point is transformed by wavelet transform, and the position of the damage cable (single-cable damage or double-cable damage) can be accurately identified.
3. Before and after the reduction in the measurement points, instead of the finite element model, the Kriging surrogate model is computed based on the deflection difference corresponding to the cable damage. Combined with the theory of the particle swarm algorithm, it is possible to accurately identify the degree of damage to the cables.
4. The proposed cable damage identification method based on monitoring data from a small number of measurement points is robust against interference and has promising applications.

**Author Contributions:** Methodology, Y.Y.; Investigation, Y.Y.; Writing—original draft preparation, Y.Y.; Software, Y.Y.; Conceptualization, M.S.; Validation, M.S.; Funding acquisition, M.S.; Writing—review and editing, M.S., Y.Y. Experiment, Y.Y. All authors have read and agreed to the published version of the manuscript.

**Funding:** This work was funded by the National Natural Science Foundations of China (Grant No. 51278315).

**Institutional Review Board Statement:** Not applicable.

**Informed Consent Statement:** Not applicable.

**Data Availability Statement:** No new data were created or analyzed in this study.

**Conflicts of Interest:** The authors declare no conflict of interest.

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
