# Peer review of "Damage Identification of Stay Cables Based on a Small Amount of Deflection Monitoring Data"

_applsci, doi:10.3390/app13095352_

Round 1

Reviewer 1 Report

The document presents a procedure for detecting damage in the cables of a cable-stayed bridge. The topic is interesting but the paper is missing some important issues and the reviewer has several points that need to be investigated/clarified before performing a detailed review.

1) English needs to be revised. In several parts the reviewer is not able to understand what the authors really mean.

2) The results are applied to a numerical case only. In real case studies, the effect of noise on measurements and model error can compromise the quality of results, especially those using derivatives of functions. In the reviewer’s opinion the authors must demonstrate the capability of the proposed method with real data or at least with -experimental data – the latter being numerical data but corrupted by noise.

Furthermore, the authors solve a two-dimensional problem (that is, assuming that the bridge is perfectly symmetrical). The article should also discuss the effect of model error, due, for example, to unintentional eccentricities of the load (producing torsion) or to an imperfect symmetry of the geometric or mechanical distribution of masses or stiffnesses. Model error can also compromise the ability of the proposed method to identify cable damage.

3) for the applicability of the proposed method to real structures, the authors have to better define which could be the experimental setup to be installed on the bridge. Indeed, it is not easy to measure the displacements of the bridge with high precision

4) the proposed procedure is based on the theory of the elastic beam on an elastic foundation. In the proposed case, the cables do not have the same stiffness due to different lengths and angles. Furthermore, the effect of the relative stiffness between cables and beam (i.e. the value of \beta) on the results has to be investigated.

5) several parts are well known to the readers of the journal, such as the theory of beams on elastic foundation, the wavelet theory and the Kriging model. These parts need to be strongly shortened.

6) it is not clear which optimization procedure is performed on the measuring point position. What is the goal of the optimization process and what procedure is applied?

Reviewer 2 Report

Reviewer’s Comments:

The manuscript “Cable Damage Identification Based on Deflection Monitoring Data of Optimized Measuring Points” is a very interesting work. In this work, aiming at the problem of cable damage identification in cable-stayed bridges, a cable damage identification method is proposed based on the deflection monitoring data of optimized measuring points. Firstly, the method of optimizing measuring points is proposed. The distribution characteristics of deflection difference before and after cable damage of cable-stayed bridge with optimized measuring points are analyzed; The first derivative of the deflection difference is transformed by wavelet transform to identify the position of the damage cable. Then, the Kriging proxy model of exponential and deflection difference is established. The objective function is constructed based on the residual deflection difference formed by the deflection difference and the measured deflection difference. While I believe this topic is of great interest to our readers, I think it needs major revision before it is ready for publication. So, I recommend this manuscript for publication with major revisions.

1. In this manuscript, the authors did not explain the importance of the Deflection Monitoring Data in the introduction part. The authors should explain the importance of Deflection Monitoring Data.

2) Title: The title of the manuscript is not impressive. It should be modified or rewritten it.

3) Correct the following statement “The results show that the position of the damaged cable can be accurately identified by the wavelet transform of the first derivative of the deflection difference of the girder before and after the damage of the cable, then the damage degree of the cable can be accurately identified by Kriging proxy model and particle swarm optimization algorithm”.

4) Keywords: The keywords should be small. So, modify the keywords.

5) Introduction part is not impressive. The references cited are very old. So, Improve it with some latest literature.

6) The authors should explain the following statement with recent references, “The formation of the subsample Zt as formation of a set of subsamples based on random selection with repetitions of objects, where some of the objects do not fall into the subsample done”.

7) Please justify the following statement “It is assumed that the damage to both cables is different (see Table 1). Figure 7 is the comparison between the deflection distribution curve obtained after the measurement point optimization and the (actual) deflection distribution curve before the measurement point optimization when the double cable is damaged in the working condition listed in Table 1”.

8) The author should provide reason about this statement “Figure 9 shows the distribution curves of the first derivative of the deflection difference of the beam before and after the double cable damage (M16 damage 15%, M7 damage 40%)”.

9) Comparison of the present results with other similar findings in the literature should be discussed in more detail. This is necessary in order to place this work together with other work in the field and to give more credibility to the present results.

10) Conclusion part is very long. Make it brief and improve by adding the results of your studies.

11) There are many grammatic mistakes. Improve the English grammar of the manuscript.

Round 2

Reviewer 1 Report

The reviewer appreciated the hard work done to improve the paper introducing several sections concerning the influence of noise on the measurements and an experimental case. They are clear and support the conclusion and the results.

The reviewer has now only one major issue and some minor points to be pointed out.

Major issue.

The optimization process of measuring point location is not yet clear. The authors declare that it is necessary to obtain the deflection data of each cable and to reduce the number of measuring points. The aim of reducing the number of measuring points is clear and the misunderstanding probably lies in the word "optimization". In the reviewer’s opinion there are 2 possibilities. 1) the authors want to perform an optimization process. In this case the objective function to be minimized (or maximized) must be defined and explained; an optimization algorithm (gradient based, genetic algorithms, Particle Swarm Optimization or others) must be applied. The number of selected points and their location should be the result of the optimization process. 2) the authors want to reduce the number of measuring points and decide to select only 14 measurement points in a preselected position. The position and the number of points must be justified or (better) it must be shown how the results change if the full set of measurement points are selected instead of the reduced sub-set - it seems what the authors present in Figures 7-9. Moreover, Figures 7-9 and Tables 3-4 clearly show that there is not much difference in deflection after or before the process. Even the authors say so.

So, in summary, why do you want to perform a measurement point optimization process? It seems unnecessary. Probably it is better to define what done a selection of a limited number of measurement points rather than an optimization procedure

Minor issues

Page 4: the notation can be simplified. Subscript i denotes the i-th cables. Each cable is placed at a allocated position x_k. SO the notation can be simplified deleting the subscript i or deleting the dependence from x_k.

Page 4: please define wi^Fi and wi^q. Even if the reviewer has understood their meaning, it is important to define all variables. Similarly, k and \gamma (see page 5) must be defined after equation 6 and or 7 not after eq. 9.

Page 5, line 170. The sentence “… Under the action of concentrated force, Deflection function Δ?(??) as unknown 170 quantity, Δ?(??) should satisfy the differential equation is derived. Solution (6) can be 171 obtained Equation (7).” Is not wel written, The reviewer suggest to simplify it as follows: “   Under the action of concentrated force, the solution of Equation (6) is”.

Page 5. The sentence “According to Equation (7), the deflection difference of the beam maximized at the anchorage point of the damaged cable and the main beam.” is unclear. Please revise it

Page 6. The sentence “…According to Equation (8) and Figure 4, when x=0, the deflection value is the largest.” Is not correct. The reviewer has understood what the authors mean but the sentence need to be revised. The abscissa X (see page 4) is the absolute coordinate, here instead is relative to each cable.

Page 6, lines 189-197. Sentences are unclear. Please rewrite.

Page 11, lines 328-330. Sentences are unclear. Please rewrite.

Page 12, line 350. Neve start a sentence with “Where…”. Use, for instance, “In Eq. y_i is…”

Page 13, line 384. Dace toolbox in Matlab. Please enter a reference.

Line 454 and 485.  “.. instead of the measured deflection difference...”  means “ instead of the reference/numercal/ exact value” ?

Line 554. “Eight damage cases” while table 4 reports only 6 cases.

Round 3

Reviewer 1 Report

The authors have made the requested changes. In my opinion, the paper can be accepted in its current form.